# Deep spectral improvement for unsupervised image instance segmentation

**Farnoosh Arefi, Amir M. Mansourian, Shohreh Kasaei** [ORCID] *

Department of Computer Engineering, Sharif University of Technology, Tehran, Iran

* kasaei@sharif.edu

## Abstract

Recently, there has been growing interest in deep spectral methods for image localization and segmentation, influenced by traditional spectral segmentation approaches. These methods reframe the image decomposition process as a graph partitioning task by extracting features using self-supervised learning and utilizing the Laplacian of the affinity matrix to obtain eigensegments. However, instance segmentation has received less attention than other tasks within the context of deep spectral methods. This paper addresses that not all channels of the feature map extracted from a self-supervised backbone contain sufficient information for instance segmentation purposes. Some channels are noisy and hinder the accuracy of the task. To overcome this issue, this paper proposes two channel reduction modules, Noise Channel Reduction (NCR) and Deviation-based Channel Reduction (DCR). The NCR retains channels with lower entropy, as they are less likely to be noisy, while DCR prunes channels with low standard deviation, as they lack sufficient information for effective instance segmentation. Furthermore, the paper demonstrates that the dot product, commonly used in deep spectral methods, is not suitable for instance segmentation due to its sensitivity to feature map values, potentially leading to incorrect instance segments. A novel similarity metric called Bray-curtis over Chebyshev (BoC) is proposed to address this issue. This metric considers the distribution of features in addition to their values, providing a more robust similarity measure for instance segmentation. Quantitative and qualitative results on the Youtube-VIS 2019 and OVIS datasets highlight the improvements achieved by the proposed channel reduction methods and using BoC instead of the conventional dot product for creating the affinity matrix. These improvements regarding mean Intersection over Union (mIoU) and extracted instance segments are observed, demonstrating enhanced instance segmentation performance. The code is available on: https://github.com/farnooshar/SpecUnIIS.

**Data Availability Statement:** To facilitate research in this area, an open-source implementation of the method (codes) and data is available at: https://github.com/farnooshar/specuniis.

## Introduction

Object segmentation, including Foregorund-Background (Fg-Bg) segmentation and instance segmentation, is a fundamental problem of computer vision with numerous applications in domains such as medical image analysis [1], autonomous driving [2, 3], and robotics [4, 5]. Despite the advancements in Deep Neural Networks (DNNs), tackling these problems still

**Funding:** The author(s) received no specific funding for this work.

**Competing interests:** The authors have declared that no competing interests exist.

presents challenges due to the requirement of dense annotation, which can be time-consuming to create. Moreover, relying on predefined class labels may limit the applicability of these methods, especially in domains such as medical image processing where expert annotation is necessary.

To overcome the limitations of supervised learning methods, alternative approaches with reduced reliance on full supervision have emerged, such as semi-supervised learning [6], weakly-supervised learning [7], and the utilization of scribbles or clicks [8–10]. Nevertheless, the aforementioned methods still face challenges as they require some form of annotations or prior knowledge of the image. On the other hand, unsupervised methods leverage self-supervised learning methods, often based on transformers, as backbones to generate attention maps that correspond to semantic segments within the input image. The extracted features from self-supervised models exhibit significant potential in aiding visual tasks, like object localization [11], object segmentation [12], semantic segmentation [13, 14], and instance segmentation [15, 16].

Recent advancements in this field have shown encouraging outcomes, mainly using unsupervised learning approaches that integrate deep features with classical graph theory for object localization and segmentation tasks. Specifically, these methods employ features extracted from a self-supervised transformer-based backbone and leverage the Laplacian of the affinity matrix and patch-wise similarities for tasks such as object localization and semantic segmentation [17, 18]. However, instance segmentation remains relatively unexplored. This task poses challenges as it entails recognizing and segmenting each individual object within an image, whereas semantic segmentation treats multiple objects of the same category as a single entity.

In this paper, we posit that some channels may contain noise and are thus not suitable for Fg-Bg segmentation, particularly for instance segmentation. To show this, we conduct an experiment using the features extracted from the Dino model [19] with the baseline method of Deep Spectral Methods (DSM) [18]. Fig 1 validates this claim by showing that certain feature channels resulting from the self-supervised backbone have reasonable accuracy in instance segmentation, as they delineate the instances based on their pixel values.

Based on this discovery, we hypothesize that identifying this specific channel, we refer to it as the promising channel, where instances are discernible through their pixel values, will improve the accuracy of instance segmentation. As depicted in Fig 2, employing the promising channel for instance segmentation yields satisfactory results across various number of instances. To find the promising channel, a Noise Channel Reduction (NCR) method is proposed, which filters out useless channels based on their entropy. This channel reduction process aims to eliminate noisy channels, thereby simplifying the creation of an affinity matrix and enhancing the results of Fg-Bg segmentation.

Furthermore, an additional channel reduction method called Deviation-based Channel Reduction (DCR) is introduced. It further eliminates specific channels based on their standard deviation. The underlying idea of this module is that specific channels in the last step are proper for Fg-Bg segmentation, as they can discriminate between foreground and background regions. However, in instance segmentation, the challenge lies in distinguishing individual object instances, noting that channels with low standard deviation do not provide sufficient information for this task.

Finally, the two-step reduced channels are employed to construct an affinity matrix. At this stage, it is found that the commonly used dot product is a sub-optimal choice for instance segmentation, as it heavily emphasizes high or low values, which are often considered to be noise. Therefore, a new metric that considers both the distribution of features and values rather than just their raw values is proposed. This addresses the limitations of the conventional dot product and leads to improved results in instance segmentation.

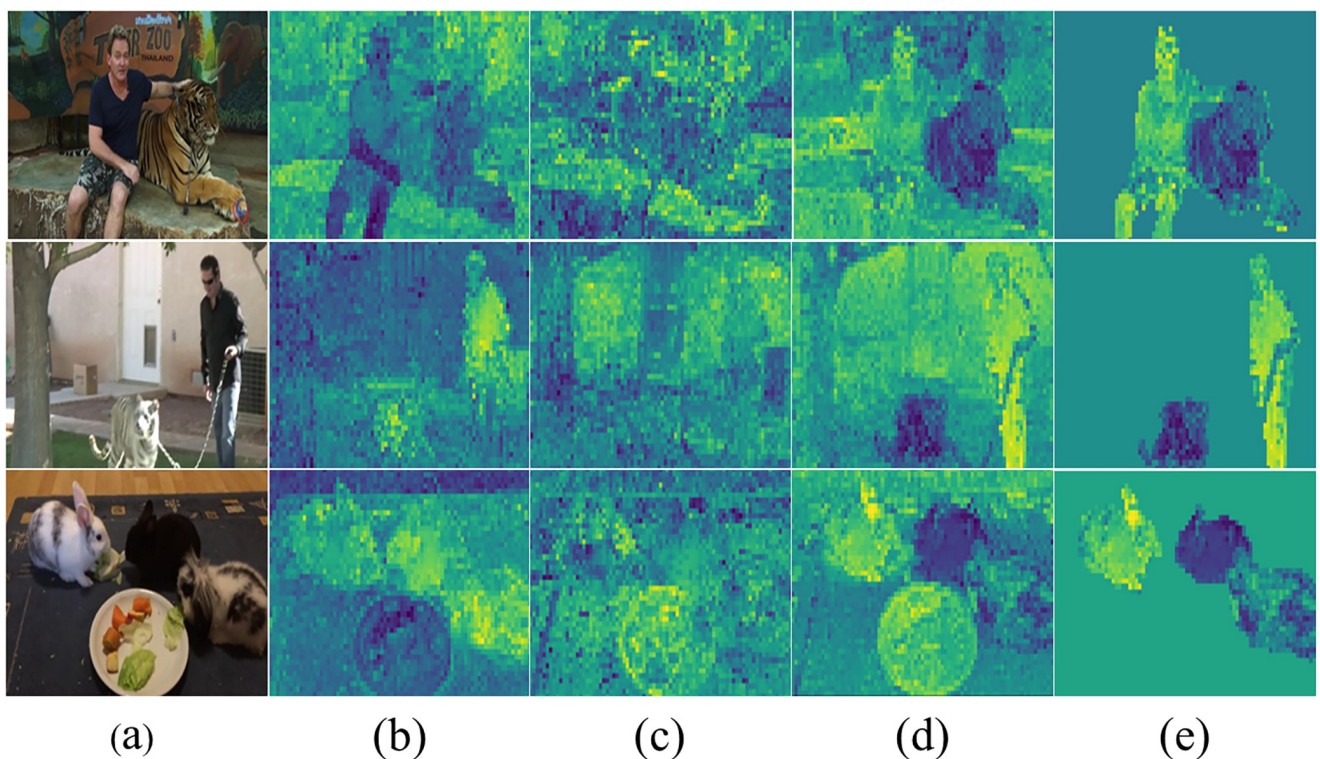

**Fig 1. Features extracted from the self-supervised backbone.** (a) Input image. (b) A channel that is suitable for foreground-background segmentation. (c) A random channel. (d) A promising channel for instance segmentation. (e) A channel suitable for instance segmentation, multiplied by the foreground mask. As shown, instances in the image are distinguishable by their pixel value.

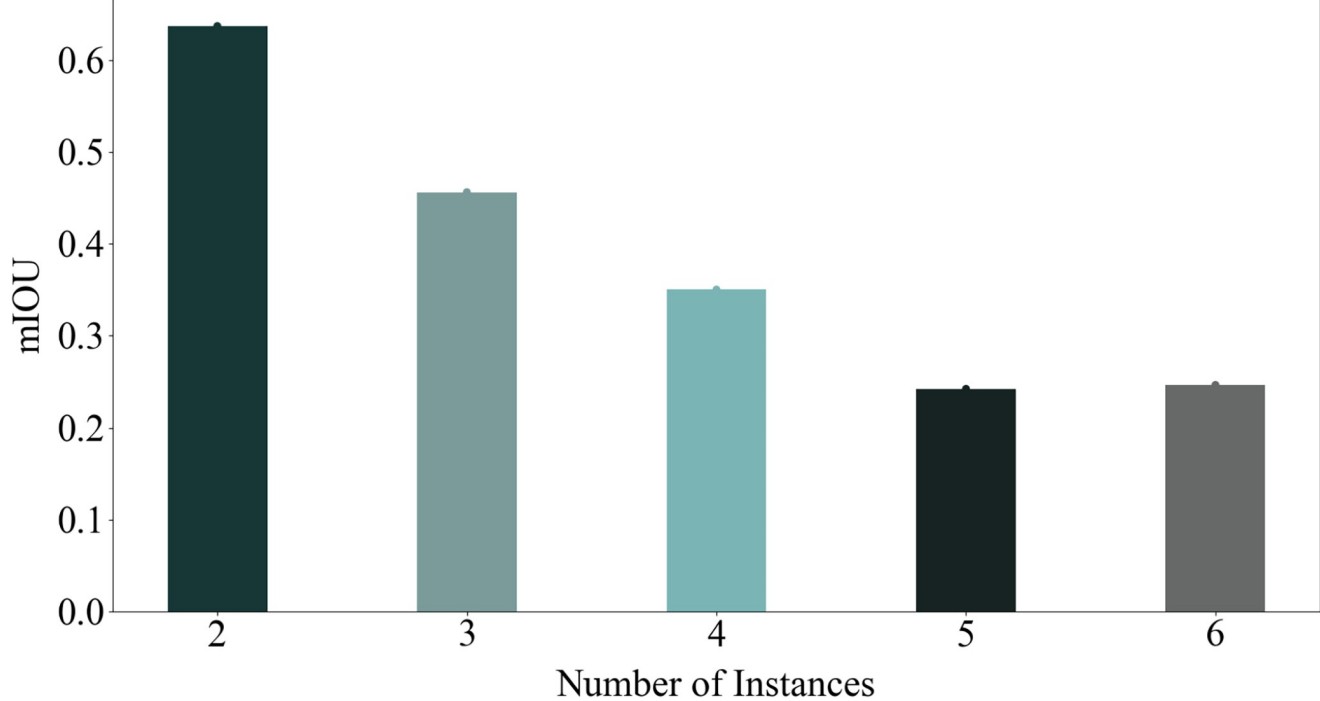

**Fig 2. Potential appropriateness of channels for intense segmentation.** As depicted in this figure, a specific channel shows promising potential for performing well in the instance segmentation task. These results were analyzed across different instances.

In summary, the contributions of this work are as follows:

- Proposing Noise Channel Reduction (NCR) method for eliminating noisy channels and achieving better Fg-Bg segmentation results.

- Deviation-based Channel Reduction (DCR) method for further reducing the data dimension while preserving the channels that are valuable for instance segmentation.

- Analyzing the limitations of the dot product for instance segmentation, and proposing a new metric based on the distribution and values of features to construct an affinity matrix suitable for the instance segmentation task.

The remaining sections of this paper are structured as follows. Section reviews the relevant literature. This is followed by a detailed explanation of the proposed method in Section. Lastly, Section provide extensive evaluations and ablation studies, as well as a discussion of the results. Finally, Section concludes the paper.

## Related work

This section includes state-of-the-art research surrounding self-supervised learning and deep spectral methods.

### Self-supervised learning

In recent years, significant advancements in self-supervised learning for visual recognition tasks have been made. Initial approaches in this field involve training a self-supervised backbone using pretext tasks such as colorization, inpainting, or rotation [20–24]. The trained backbone is then applied to the target task. More recently, Contrastive Learning has been employed to generate similar embeddings for different augmentations of the image while generate different embeddings for different images [25–28]. However, some approaches have aimed to avoid reliance on negative samples [29, 30] or have utilized clustering techniques [31, 32].

Inspired by the Bidirectional Encoder Representations from Transformers (BERT) [33], various methods have been proposed that train models by reconstructing masked tokens [34–36]. For instance, Masked Autoencoder (MAE) [36] takes input images with a high ratio of masked pixels and employs an encoder-decoder architecture to reconstruct the missing pixels.

There has been a surge of attention in the context of self-supervised learning with vision transformers. Momentum Contrast (MoCo-v3) [37], for instance, has achieved impressive results by employing contrastive learning on vision transformers. On the other hand, the self-Distillation with NO labels (DINO) [19] has introduced a self-distillation approach for training vision transformers, creating features explicitly beneficial for image segmentation.

This study employs DINO as a self-supervised backbone, leveraging its feature maps to demonstrate efficacy in image segmentation. By incorporating deep spectral methods, the study accomplishes foreground-background separation and instance segmentation.

### Spectral methods

The concept of spectral graph theory was initially introduced in [38]. Subsequent research focused on the discrete formulation of graphs, establishing a connection between global graph features and the eigenvalues/eigenvectors of their Laplacian matrix [39, 40]. The work in [39] proposed that the second smallest eigenvalue of a graph, known as the Fidler eigenvalue, serves as a measure of graph connectivity. Additionally, [40] demonstrated that the eigenvectors of graph Laplacians can be utilized to achieve graph partitions with minimum energy. However,

with machine learning and computer vision advancements, [41, 42] introduced the pioneering methods for image segmentation through spectral clustering.

Following DSM [18], numerous works have been presented that primarily focus on object localization [17, 43, 44], semantic segmentation [13, 45, 46], and video object segmentation [12, 47, 48]. The work in [49] adopts a novel approach by utilizing DINO along with image and flow features as inputs. They construct a fully connected graph based on image patches and employ graph-cut techniques to generate binary segmentation. The generated masks are then utilized as pseudo-labels to train a segmentation network. Additionally, [48] employs the Ncut algorithm [41] to perform segmentation using the same similarity matrix derived from image patches. They further employ the Fidler eigenvector to bi-partition the graph, followed by a refinement step for video segmentation.

However, deep spectral methods have generally received less attention in instance segmentation than object localization and semantic segmentation. To address this gap, [43] introduces MaskCut, a method capable of extracting masks for multiple objects within an image without relying on supervision. Similar to [48], MaskCut constructs a similarity matrix on a patch-wise basis using a self-supervised backbone. The Normalized Cuts algorithm is then applied to this matrix, yielding a single foreground mask for the image. Subsequently, the affinity matrix values corresponding to the foreground mask are masked, and the process is repeated to discover masks for other objects. Another recent method, proposed by [15], investigates the features extracted from multiple self-supervised transformer-based backbones. That approach employs two different feature extractors. Initially, a new feature extractor is utilized, followed by spectral clustering with varying numbers of clusters. Simultaneously, DINO is employed along with spectral clustering using two clusters to generate a foreground mask. Finally, candidate masks from the previous step that exhibit significant intersection with the foreground mask are selected as the final masks.

In this work, starting from [18], an improvement over the deep spectral methods for instance segmentation is proposed. Through the analysis conducted in this study, it is demonstrated that not all channels of the feature maps obtained from DINO are useful for effective instance segmentation. As a result, two steps of channel reduction are proposed to enhance the overall segmentation performance. Furthermore, the study reveals that the dot product is unsuitable for creating the affinity matrix in instance segmentation. To address this issue, a new metric is proposed that is specifically tailored for generating the affinity matrix for instance segmentation.

## Proposed method

This section starts with a concise introduction to the Deep Spectral Method (DSM) [18], serving as the baseline for this work. Following that, the two channel reduction methods are elaborated upon, providing an explanation of the underlying concepts behind each method. Subsequently, a novel similarity metric specifically designed for instance segmentation is proposed, replacing the conventional dot product. The overall framework of instance segmentation is then outlined, providing a comprehensive understanding of the approach.

### Preliminary

Consider a weighted undirected graph, denoted by $G = (V, E)$, with its corresponding adjacency matrix $W = \{w(u, v):(u, v) \in E\}$. This graph's Laplacian matrix $L$ can be defined as $L = D - W$. Here, $D$ represents a diagonal matrix with entries being the row-wise sums of $W$. In spectral graph theory, the eigenvectors and eigenvalues of $L$ hold significant data. The eigenvectors, denoted as $y_i$, correspond to the eigenvalues $\lambda_i$ and form an orthogonal basis that

allows for the smoothest function representation on $G$. Hence, it is natural to express functions defined on $G$ using the graph Laplacian's eigenvectors.

In classical image segmentation, $G$ can be interpreted as the pixels of an image $I \in \mathbb{R}^{HW}$, where $H$ and $W$ represent the image's dimensions. The edge weights $W \in \mathbb{R}^{HW \times HW}$ correspond to the affinities or similarities between pairs of pixels. The eigenvectors $y \in \mathbb{R}^{HW}$ can be interpreted as representing soft image segments, providing a way to group pixels into coherent regions.

Graph partitions that divide the image into disjoint segments are often called graph cuts. In this context, the value of a cut between two partitions reflects the total weight of edges removed by the partitioning process. A normalized version of graph cuts, as described in [41], emerges naturally from the eigenvectors of the normalized Laplacian. In this case, achieving optimal bi-partitioning becomes equivalent to identifying the appropriate eigenvectors of the Laplacian.

Recently, [18] proposed a method that combines the explained background with deep learning. Fig 3(a) illustrates the pipeline of [18] approach. It utilizes feature maps extracted from a self-supervised backbone, denoted as $F$, to create a patch-wise affinity matrix $W$. From $W$, it extracts the eigenvectors of its Laplacian, $L$, which enables the decomposition of an image into soft segments: $\{y_0, \ldots, y_{n-1}\} = eigs(L)$. These eigensegments are then utilized for both Fg-Bg segmentation and semantic segmentation. For more details regarding this method, please refer to [18].

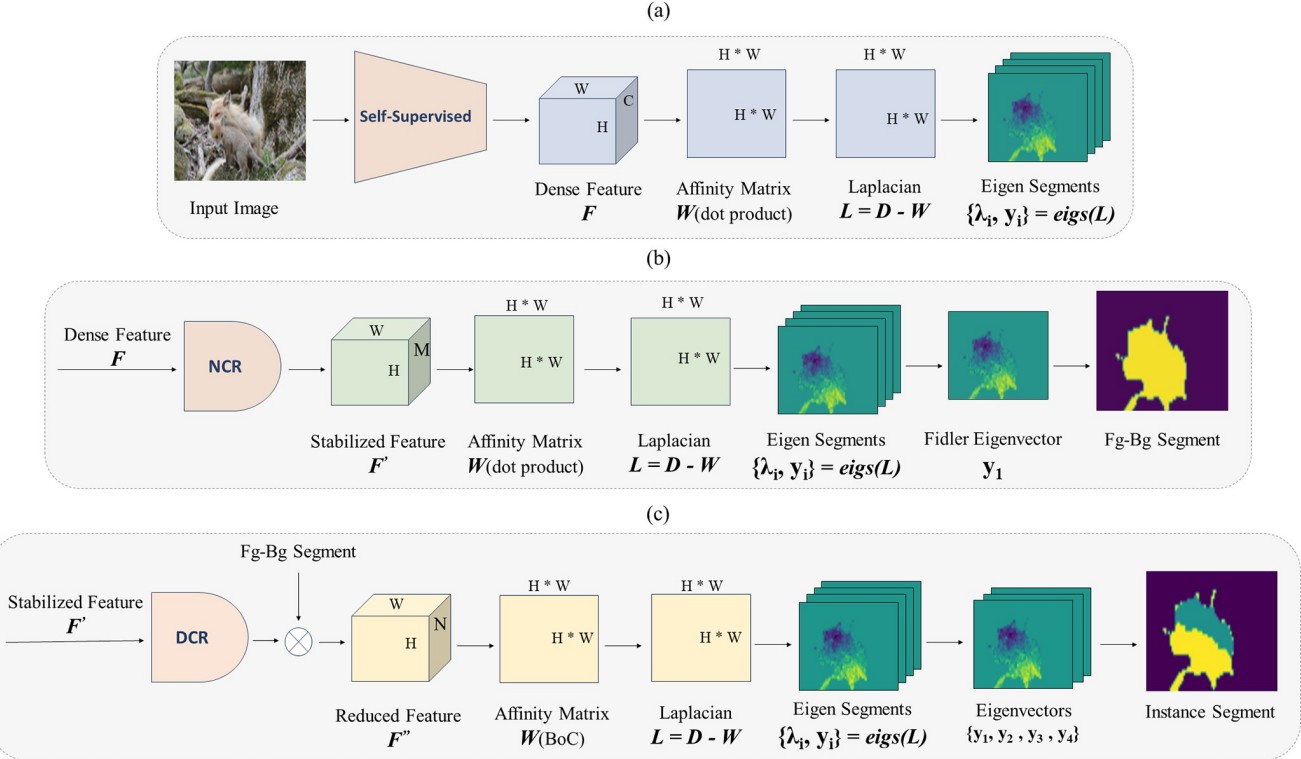

**Fig 3. Pipelines of discussed deep spectral methods.** (a) Workflow of [18]. An affinity matrix is created using a dot product with features from a self-supervised backbone. Eigenvectors of the Laplacian matrix derived from the affinity matrix are utilized for segmentation tasks. (b) Application of the proposed NCR module on the features from the self-supervised backbone to remove noisy channels. The Fiedler eigenvector is then employed for foreground-background segmentation. (c) Pipeline for instance segmentation. Stable feature map channels are further reduced based on their standard deviation to enhance feature richness. The resulting feature map is multiplied by the foreground mask, and the affinity matrix is created using the BoC metric. Finally, pixels are clustered using the eigenvectors of the Laplacian matrix, resulting in instance segmentation.

## Noise Channel Reduction (NCR)

Not all feature maps extracted from a self-supervised backbone are suitable for segmentation. Some of them may contain noise, which can negatively impact the creation of the affinity matrix. To address this issue, a channel pruning method is proposed to reduce the number of channels based on their entropy. In this context, instability refers to the presence of unrelated or irrelevant information in the channels for the target task, compared to more stable features that contain rich information with few irrelevant components. Finding the most stable feature map is challenging as it heavily depends on the specific task or dataset. However, this study aims to move towards a more stable feature map. To achieve this, the entropy of the channels is utilized to eliminate channels with less informative content. The entropy of a channel quantifies the level of disorder or randomness within the channel. Given a feature map, the probability distribution function (or the histogram) is first calculated for each channel as follows:

$$PDF(c) = \frac{Hist(c)}{H \times W}; \forall c \in C, \tag{1}$$

where $Hist(c)$ denotes the histogram of the $c$-th channel, while $H$, $W$, and $C$ represent the height, width, and the number of channels of a feature map, respectively. The entropy of a channel is then defined as:

$$Entropy(c) = -\sum_{b=1}^{B} PDF^b(c).log_2(PDF^b(c)), \tag{2}$$

where $PDF^b(c)$ represents the probability of the $b$-th bin in the channel's histogram $c$ and $B$ is equal to the total number of bins, which is considered equal to 30 in this paper. The channels of the feature map $F$ are then sorted based on their entropy. The first $M$ channels with the lowest entropy are retained, where $M$ is a hyper parameter. This selected subset is denoted as $F' \in \mathbb{R}^{H \times W \times M}$, which represents a more stable version of $F$.

Fig 4 illustrates various channels of an input image, each exhibiting different entropy levels. It demonstrates that channels with higher entropy tend to contain more noise, while channels with lower entropy exhibit semantically richer information for distinguishing between Fg-Bg regions. The proposed NCR module reduces the number of channels, resulting in improved feature maps for Fg-Bg segmentation.

Finally, the Fg-Bg segmentation task is performed, as depicted in Fig 3(b). Initially, the features extracted from the self-supervised backbone, denoted by $F$, pass through the NCR module, resulting in more stabilized feature maps, $F'$. Subsequently, the affinity matrix is constructed by taking the dot product between $F'$ and its transpose. Then the Laplacian matrix is computed, and the second smallest eigenvector (commonly referred to as the Fiedler eigenvector) is utilized for Fg-Bg segmentation.

## Deviation-based Channel Reduction (DCR)

As illustrated in Fig 1, specific channels exhibit more distinguishable instances, indicating their potential usefulness in instance segmentation tasks. The purpose of designing a DCR module is to select channels that maximize the separation between instances. To address this, channels are sorted based on their standard deviation (STD). A channel with a lower STD tends to have less variation between instances, making it less suitable for instance segmentation. It is important to note that a higher STD does not necessarily guarantee better performance for instance segmentation, as noise can also contribute to increased STD. However, since the NCR module likely removed channels with noise, using channels with a higher

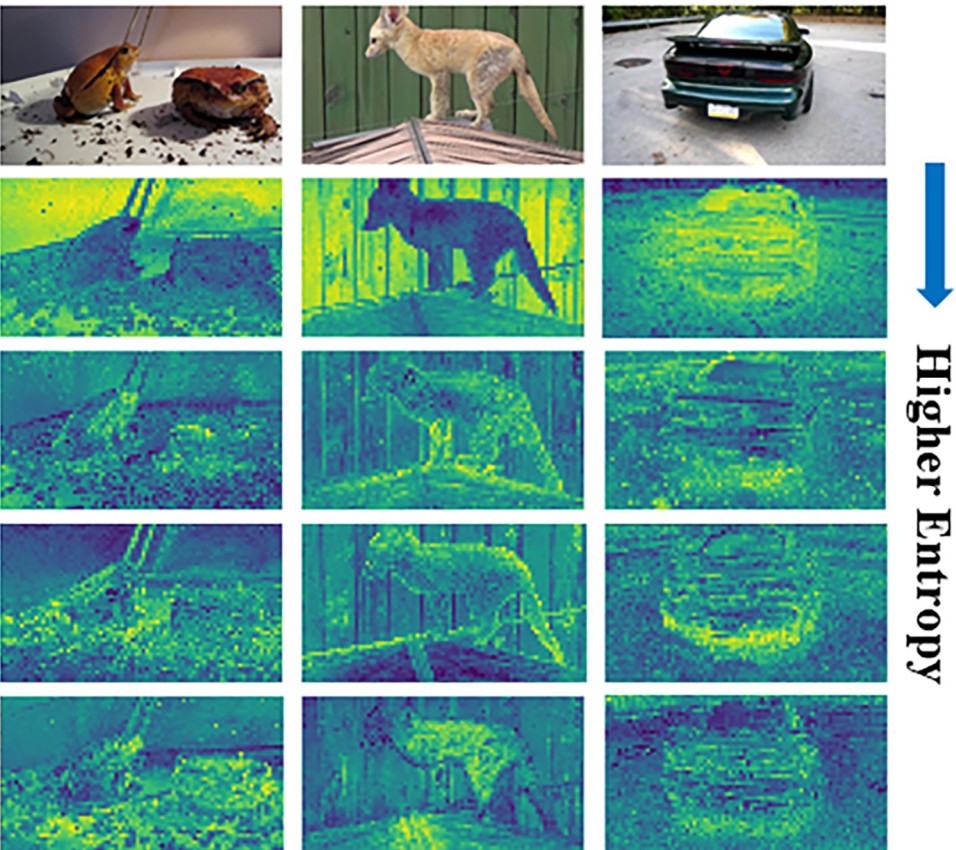

**Fig 4. Visualization of some channels from the self-supervised backbone.** As evident in this figure, lower entropy corresponds to a better representation of objects in images, while higher entropy results in a more unclear representation resembling noise.

standard deviation (STD) will be useful for instance segmentation. Therefore, selecting channels based on their STD is a reasonable choice. To accomplish this, the STD of each channel is calculated using the following formula:

$$STD(c) = \sqrt{\frac{1}{H \times W} \sum_{x=1}^{H \times W} (x - \bar{x})^2}; \forall c \in F', \tag{3}$$

where $\bar{x}$ is the mean of the values in a specific channel. Then, all channels are sorted based on their STD values, and the first $N$ channels with the highest STD are retained, where the $N$ is a hyper parameter. This process results in a final feature map denoted by $F'' \in \mathbb{R}^{H \times W \times N}$. Fig 5 demonstrates the impact of DCR with N = 60. As indicated by the orange curve, the standard deviation of the channels decreases, and simultaneously, $\Delta$, representing the average difference between instances, also decreases. Therefore, there is a strong likelihood that channels with a higher standard deviation will exhibit a greater average difference between instances.

## Deep spectral instance segmentation

**Dot product for instance segmentation.** As shown in Fig 3(a), [18] utilizes the dot product to create an affinity matrix, which is subsequently employed in downstream tasks such as

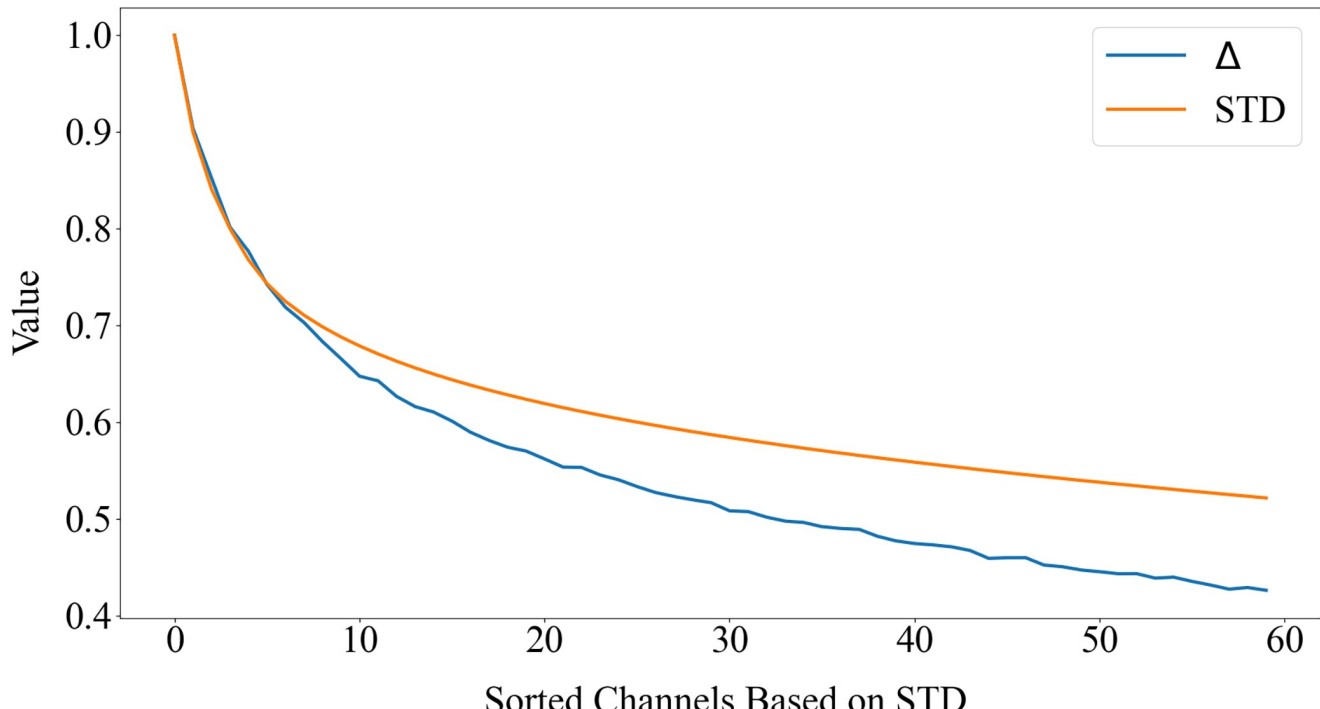

**Fig 5. Influence of DCR on the distinction between instances in YouTube-VIS 2019 dataset.** As depicted by the orange curve, the standard deviation of the channels diminishes while simultaneously, $\Delta$, which represents the average difference between instances, also decreases. Consequently, channels with a higher standard deviation will probably display a more significant average difference between instances.

Fg-Bg segmentation or semantic segmentation. However, in this section, we challenge the suitability of the dot product as a proper method for generating the affinity matrix, specifically for instance segmentation. This claim is supported by two reasons:

- **Sensitivity to the feature vectors values**: The dot product is highly sensitive to the values in the feature maps. If there are extreme values, either very high or very low (referred to as irregular values), within a feature vector, they can significantly influence the affinity matrix even after normalization. Fig 6 illustrates an instance where one of the channels in the feature maps contains irregular values. Consequently, these values heavily impact the final affinity matrix, resulting in their segmentation as a separate region.
While the dot product can be useful for Fg-Bg segmentation, as it considers the significant differences in pixel values, especially along edges, it may not be as effective for instance segmentation. In instance segmentation, the focus is on capturing more than just the values of the pixels; additional details are required to segment different instances accurately.

- **Negligence of the feature distribution**: In image instance segmentation, it is crucial to consider the distribution of features rather than solely relying on their exact values. Pixels with similar feature distributions should be segmented as the same instances. As depicted in Fig 6, there are irregular values present in some of the channels. It is essential to note that these seemingly irregular patterns are part of the same instance. The problem with using dot product is that it treats these regions as separate instances, disregarding the underlying feature distribution.
An ideal affinity matrix would possess the property that pixels of the same instance exhibit similar feature distributions while having maximum dissimilarity with pixels from other

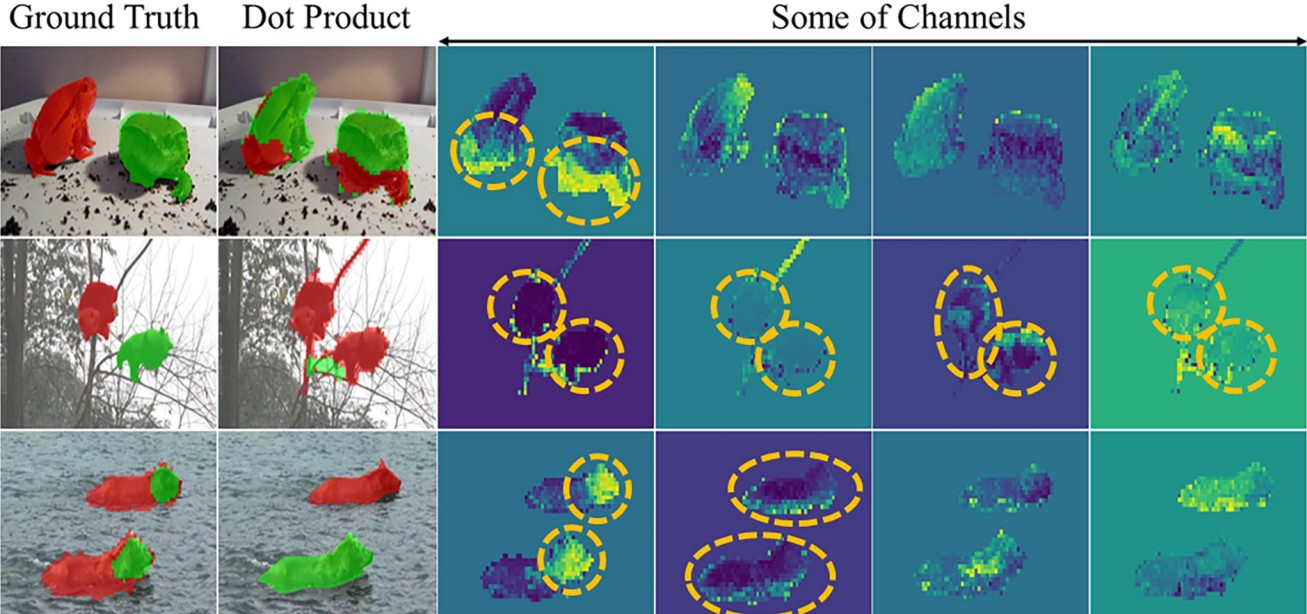

**Fig 6. Qualitative results for instance segmentation, when using dot product to create the affinity matrix.** As illustrated, in some channels from the feature map, there are pixels with very high or low values. Using dot product sensitive to these values can lead to incorrect instance segmentation outputs.

instances. To achieve this, a metric that considers the feature distribution is necessary. Additionally, for effective comparison, normalization should be carried out using a similarity metric sensitive to the values in the feature maps.

**Capturing feature distribution.** We propose using the Bray-Curtis similarity metric to capture information about the distribution of features. Unlike metrics that rely solely on exact values, the Bray-Curtis metric emphasizes distribution. Originally developed for ecological or community data analysis [50], this metric has also found valuable applications in machine learning tasks. The Bray-Curtis dissimilarity and similarity between two feature vectors, $U$ and $T$, are defined as follows:

$$BC_{diss} = \frac{\sum |u_i - t_i|}{\sum |u_i + t_i|}, \qquad (4)$$

$$BC_{sim} = \frac{1}{1 + BC_{diss}}. \qquad (5)$$

The Bray-Curtis similarity metric can effectively capture instances with similar patterns but different values in the affinity matrix. This can be better understood by examining Fig 7, which highlights the difference between the Bray-Curtis similarity ($BC_{sim}$) and the dot product. This example depicts three pixels of the same instance, characterized by nearly identical feature distributions. Some channels contain randomly added or subtracted high values to the original values. When employing the dot product, the similarity matrix, which reflects the resemblance between these three pixels, contains distinct values. In contrast, the three pixels exhibit significant closeness when utilizing the Bray-Curtis similarity metric.

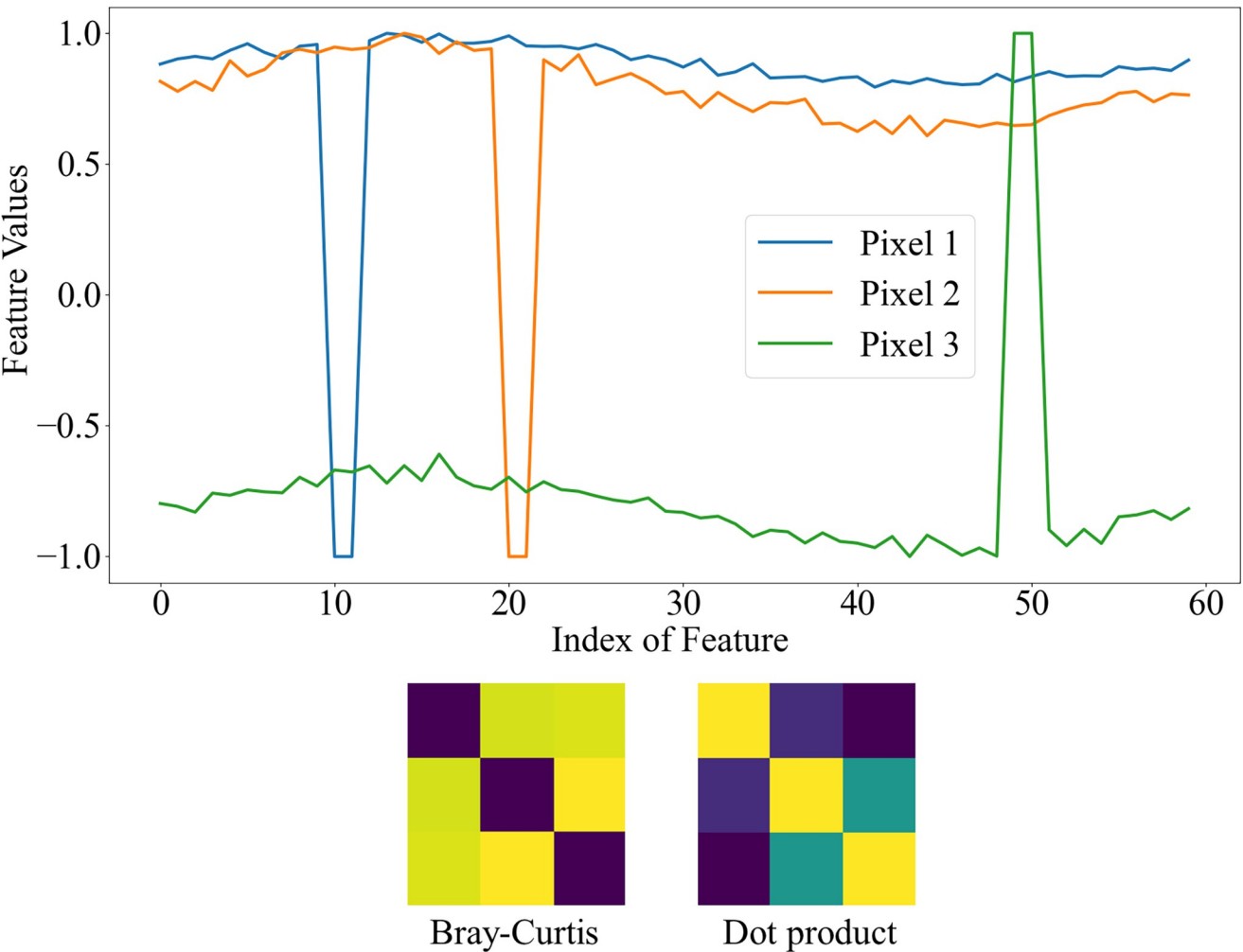

**Fig 7. Comparison of dot product and Bray-Curtis metrics in creating the affinity matrix.** Three pixels belonging to the same instance are selected from the feature maps, and random noise is added to some of their channels. The affinity matrix created by the dot product is unsuitable for our purpose. In contrast, the matrix created by the Bray-Curtis matrix correctly demonstrates the correlation between the three pixels.

**Affinity matrix creation.**　As mentioned previously, when constructing the affinity matrix, considering the distribution of features, it is crucial to consider a criterion for the similarity of pixels. However, when examining the distribution of features, less emphasis is placed on the values. To prioritize the feature values, the similarity of two feature vectors can be equated to the inverse of their maximum distance, known as the Chebyshev distance. The Chebyshev dissimilarity ($CH_{diss}$) and the Chebyshev similarity criterion ($CH_{sim}$) between two feature vectors, $U$ and $T$ are then defined as follows:

$$CH_{diss} = max_i(|u_i - t_i|), \tag{6}$$

$$CH_{sim} = \frac{1}{1 + CH_{diss}}. \tag{7}$$

Concerning the two proposed criteria, the Braycurtis over Chebyshev ($BoC$) metric is defined for constructing the affinity matrix, derived from the ratio of Bray Curtis and

Chebyshev criteria:

$$BoC = \frac{BC_{sim}}{CH_{sim}}. \tag{8}$$

Utilizing the Chebyshev similarity alone may result in the inclusion of non-significant regions. This is because giving maximum attention to the difference between two feature vectors increases the probability of capturing irrelevant information. However, when considering the feature distribution, we can apply a penalty value proportional to the maximum difference between the two vectors by utilizing the Chebyshev distance. If there is a small Chebyshev distance between two vectors, the similarity between the vertices reflects the similarity of the feature distribution. However, the larger this distance, the greater the penalty applied to this feature distribution.

**Instance segmentation paradigm.** The final framework for instance segmentation is illustrated in Fig 3(c). Stabilized features, denoted as $F'$, are fed into the proposed DCR module, where channels with higher standard deviation are retained, resulting in $F''$. These selected channels are then multiplied by the foreground mask, and an affinity matrix is created using the proposed BoC similarity metric. In the last step, the first four small eigensegments, excluding $y_0$ (equivalent to noise), are utilized for instance segmentation. To achieve this, a clustering algorithm with an appropriate number of classes is applied to the eigensegments, resulting in the extraction of instance masks.

## Experimental results

This section introduces the datasets, evaluation metrics, and details on implementation. Then, experimental results and ablation studies are discussed to validate the proposed method.

### Dataset and evaluation metrics

For the Fg-Bg segmentation task, three datasets are utilized: YouTube-VIS 2019 (train) [51], PascalVOC 2012 (train/validation) [52], and Davis 2016 [53]. These datasets comprise 61,845, 2,913, and 3,455 images, respectively.

Given our focus on instance segmentation, we specifically select images containing multiple instances for our analysis. In this section, we utilize both the YouTube-VIS 2019 and OVIS [54] datasets. To ensure the quality of our analysis, we exclude images based on two criteria: firstly, if the size of their objects is less than 0.07 times the image size, and secondly, if the ratio of the smallest instance to the largest instance is less than 0.3. Moreover, OVIS often exhibits significant occlusion, particularly in small dimensions, which can degrade the quality of test images. Therefore, we only test images with an occlusion level below 0.5 according to the MBOR [54] criterion. As a result, the final YouTube-VIS 2019 dataset comprises 10,285 images, and OVIS comprises 1,596 images that meet these criteria and are thus utilized as the test set.

The F-score metric is employed to evaluate the Fg-Bg segmentation task. The F-score combines precision and recall to provide a balanced measure of the model's performance. In binary segmentation, the objective is to classify each pixel or region in an image as either foreground or background. Precision measures the proportion of correctly classified foreground pixels out of all the pixels predicted as foreground. Recall quantifies the proportion of correctly classified foreground pixels out of all the actual foreground pixels in the image.

For the instance segmentation task, a linear assignment is performed using the Hungarian algorithm to determine the correspondence between predictions and instances. The average mIoU of all instances is then reported. The mIoU measures the intersection of the prediction

and ground truth masks divided by their union, providing an overall evaluation of the instance segmentation performance.

## Implementation details

After extracting the mask using the proposed method for Fg-Bg segmentation, a post-processing step is performed. To eliminate small regions, a median kernel with a size of 5x5 is applied to the mask. Additionally, in cases where the foreground and background are inversely predicted, the foreground and background labels are swapped while taking into account the boundaries according to DINO [19], similar to [18]. The backbone employed in the upcoming tests is ViT-s16 [19].

## Segmentation results

**Fg-Bg segmentation.** To validate the effectiveness of the proposed NCR module, experiments were conducted to evaluate the Fg-Bg segmentation results for different values of $M$, the number of remaining channels after NCR. Fig 8 illustrates the impact of stabilizing features with varying values of $M$, both with and without post-processing, on the Youtube-VIS 2019 dataset. The results indicate that preserving 1/3 of the channels (M = 1/3 C) with the lowest entropy yields the best F-score for this task. Fig 9 presents similar results from the PascalVOC 2012 dataset, where M = 1/5C leads to the best performance. Table 1 provides quantitative numbers for different values of $M$ for some datasets. It demonstrates that, for the Youtube-VIS 2019 dataset, retaining nearly 1/3 of the channels with the lowest entropy can improve segmentation results by 2% with post-processing and 3% without post-processing. Also, in the Pascal VOC 2012 dataset, retaining 1/5 of the channels increases accuracy by about 1%, and in the DAVIS 2016 dataset, retaining 1/3 of the channels increases accuracy by 5%. Notably, in

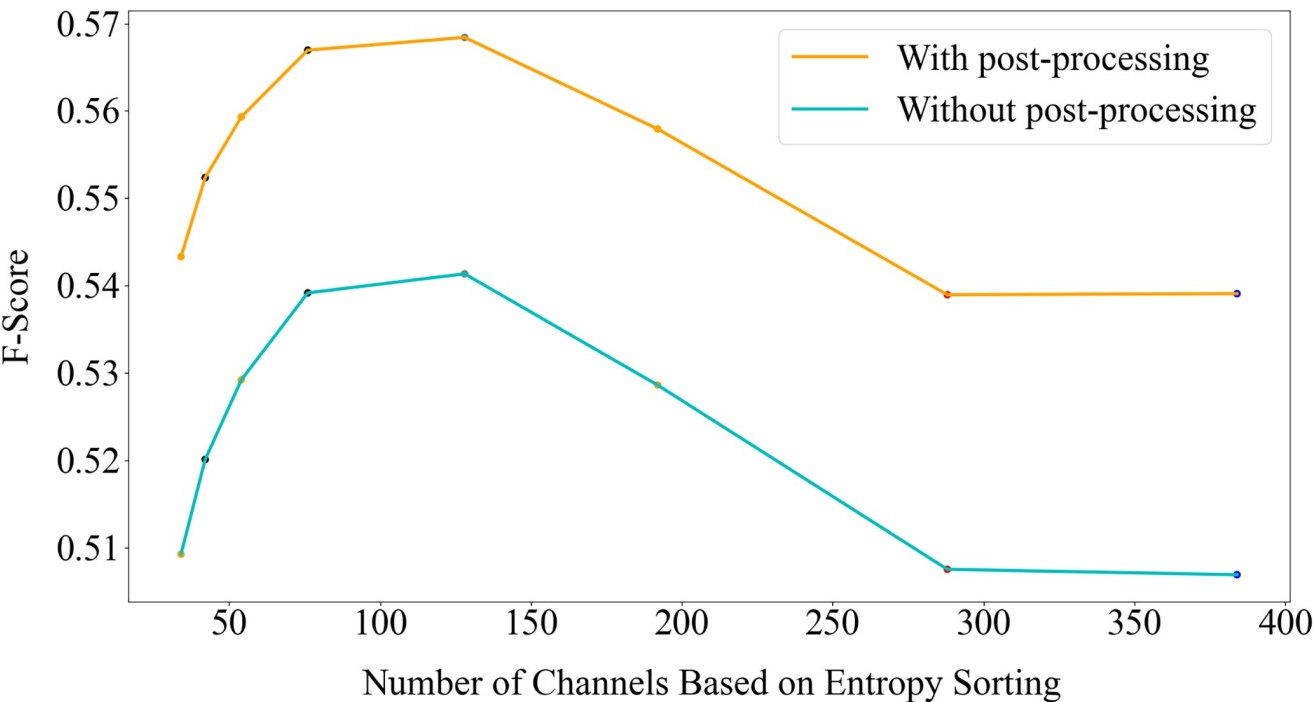

**Fig 8. Evaluation of the NCR module on the YouTube-VIS 2019 dataset.** Results of Fg-Bg segmentation for various values of $M$. Channels are sorted in ascending order based on their entropy.

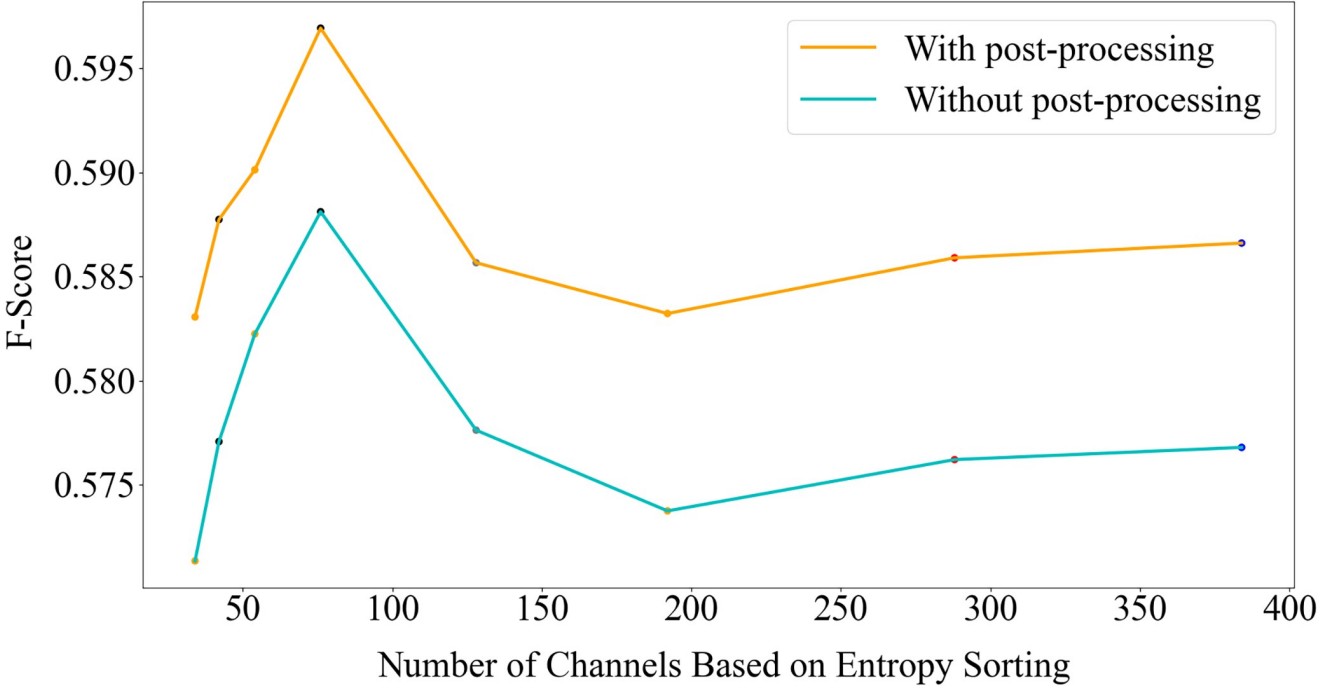

**Fig 9. Evaluation of the NCR module on the PascalVOC 2012 dataset.** Results of Fg-Bg segmentation for different values of *M*. Channels are arranged in ascending order based on their entropy.

the Youtube-VIS 2019 dataset even retaining only 20% of the channels can yield better results than using all the channels, validating that self-supervised learning can introduce noisy channels, and not all channels contribute to effective segmentation.

The findings suggest that reducing channels based on their entropy results in more stable feature maps, which enhances segmentation performance. However, pruning too many channels also leads to decreased results, indicating that useful channels from the backbone network have been discarded. Fig 10 provides visual examples illustrating the impact of stabilizing the feature map on segmentation results. It demonstrates that a more stable feature map leads to improved segmentation quality, as unrelated objects belonging to the background are correctly identified and not considered foreground.

**Instance segmentation.** Table 2 presents the mIoU results for instance segmentation using various metrics for creating the affinity matrix. Cosine, correlation, L1, L2, and Mahalanobis metrics are reversed, akin to the relationships in Eqs 5 and 7, to demonstrate similarity. NCR and DCR with $M$=128 and $N$=60, the number of remaining channels after NCR and DCR respectively, are utilized in all Tables 2–4. The results indicate that the proposed metric, BoC, achieves the best performance, with approximately 2% higher mIoU on the YouTube-VIS 2019 dataset and 2.8% higher mIoU on the OVIS dataset compared to the dot product metric used in [18]. The proposed metric demonstrates robustness to the values of feature vectors and performs well in tasks where the value alone is insufficient for discriminating the data, such as instance segmentation. In contrast, metrics like cosine similarity, correlation, L1 and L2 distances are more sensitive to the values of feature vectors.

Another advantage of the proposed metric is its performance in occlusion scenarios. It effectively highlights differences and is less susceptible to the effects of occlusion. Table 3 showcases the instance segmentation results under various levels of occlusion. The Mean

**Table 1. F-score results for Fg-Bg segmentation, considering different values of M, with and without post-processing.**

| Youtube-VIS 2019 | | |
|---|---|---|
| **Value of M** | **w post-processing** | **w/o post-processing** |
| M = C | 53.9 | 50.68 |
| M = 3C/4 | 53.89 | 50.74 |
| M = C/2 | 55.79 | 52.85 |
| M = C/3 | **56.84** | **54.13** |
| M = C/5 | 56.69 | 53.91 |
| M = C/7 | 55.93 | 52.91 |
| M = C/9 | 55.23 | 52 |
| M = C/11 | 54.33 | 50.92 |
| **PascalVoc 2012** | | |
| **Value of M** | **w post-processing** | **w/o post-processing** |
| M = C | 58.66 | 57.68 |
| M = 3C/4 | 58.59 | 57.62 |
| M = C/2 | 58.32 | 57.37 |
| M = C/3 | 58.56 | 57.76 |
| M = C/5 | **59.69** | **58.81** |
| M = C/7 | 59.01 | 58.22 |
| M = C/9 | 58.77 | 57.70 |
| M = C/11 | 58.30 | 57.13 |
| **Davis 2016** | | |
| **Value of M** | **w post-processing** | **w/o post-processing** |
| M = C | 50.68 | 46.63 |
| M = 3C/4 | 51.51 | 47.41 |
| M = C/2 | 54.77 | 50.88 |
| M = C/3 | **55.89** | **52.00** |
| M = C/5 | 55.32 | 51.07 |
| M = C/7 | 54.21 | 46.61 |
| M = C/9 | 53.23 | 48.15 |
| M = C/11 | 51.90 | 46.56 |

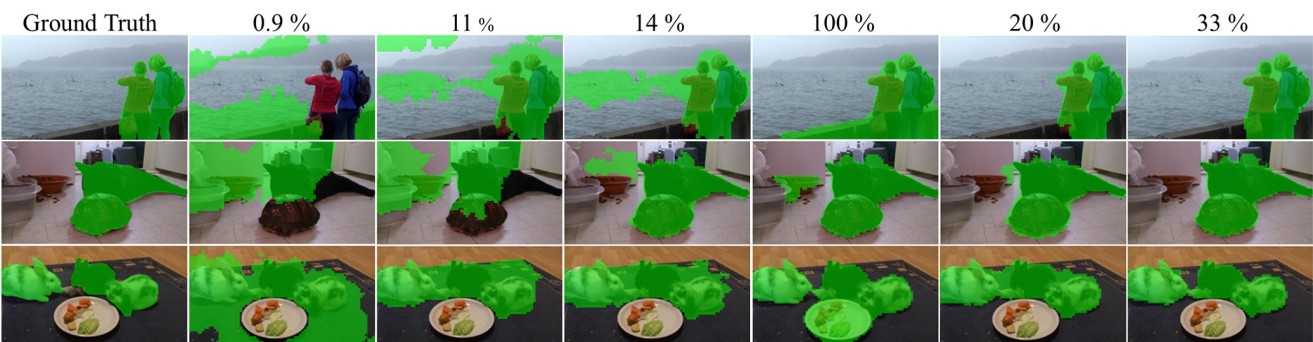

**Fig 10. Qualitative outcomes of Fg-Bg segmentation on Youtube-VIS 2019 dataset.** As illustrated, Percentages indicate the proportion of channels preserved from NCR. Precise number of channels to be retained varies for each dataset. It is determined during the generation of final masks.

**Table 2. A comparison of instance segmentation results between different metrics for creating the affinity matrix, with the proposed metric, regarding mIoU on the Youtube-VIS 2019 and OVIS datasets.**

| Metric mIoU (%) | Youtube-VIS 2019 | OVIS mIoU (%) |
|---|---|---|
| Mahalanobis | 25.27 | 25.91 |
| L1 | 31.53 | 33.74 |
| Dot product | 32.71 | 33.34 |
| L2 | 32.77 | 34.92 |
| Chebyshev | 33.09 | 34.63 |
| Cosine | 33.56 | 35.57 |
| Correlation | 34.08 | 35.93 |
| Braycurtis | 34.14 | 35.99 |
| **BoC** | **34.41** | **36.14** |

**Table 3. Quantitative results for different metrics under varying levels of occlusion in terms of mIoU on the Youtube-VIS 2019 dataset.**

| Metric | MBOR 0.01–0.14 | MBOR 0.14–0.34 | MBOR $\geq$0.34 |
|---|---|---|---|
| Mahalanobis | 24.46 | 27.20 | 26.86 |
| L1 | 31.92 | 34.18 | 30.25 |
| Dot product | 33.50 | 35.25 | 30.69 |
| L2 | 33.42 | 35.26 | 30.99 |
| Chebyshev | 33.65 | 35.91 | 31.31 |
| Cosine | 34.03 | 36.26 | 31.38 |
| Correlation | 34.76 | 36.67 | 31.64 |
| Braycurtis | 34.67 | 37.06 | 31.88 |
| **BoC** | **34.94** | **37.38** | **32.33** |

**Table 4. Quantitative results for instance segmentation considering different ratio values, in terms of mIoU on the Youtube-VIS 2019 dataset.**

| Metric | Ratio 1.26–1.64 | Ratio $\geq$1.64 |
|---|---|---|
| Mahalanobis | 26.57 | 23.98 |
| L1 | 32.10 | 30.97 |
| Dot product | 34.23 | 31.20 |
| L2 | 33.54 | 31.99 |
| Chebyshev | 34.08 | 32.10 |
| Cosine | 34.23 | 32.90 |
| Correlation | 34.98 | 33.18 |
| Braycurtis | 34.91 | 33.38 |
| **BoC** | **35.10** | **33.71** |

Boundary Overlap Ratio (MBOR) introduced by [54] indicates the degree of occlusion, with a higher MBOR value indicating more severe occlusion. It can be observed that the proposed metric performs better in scenarios with heavy occlusion. Fig 11 provides a qualitative comparison of different metrics for varying MBOR values.

Furthermore, when two instances in an image have different sizes, it is common for the smaller object to be considered as part of the larger object. However, the proposed metric

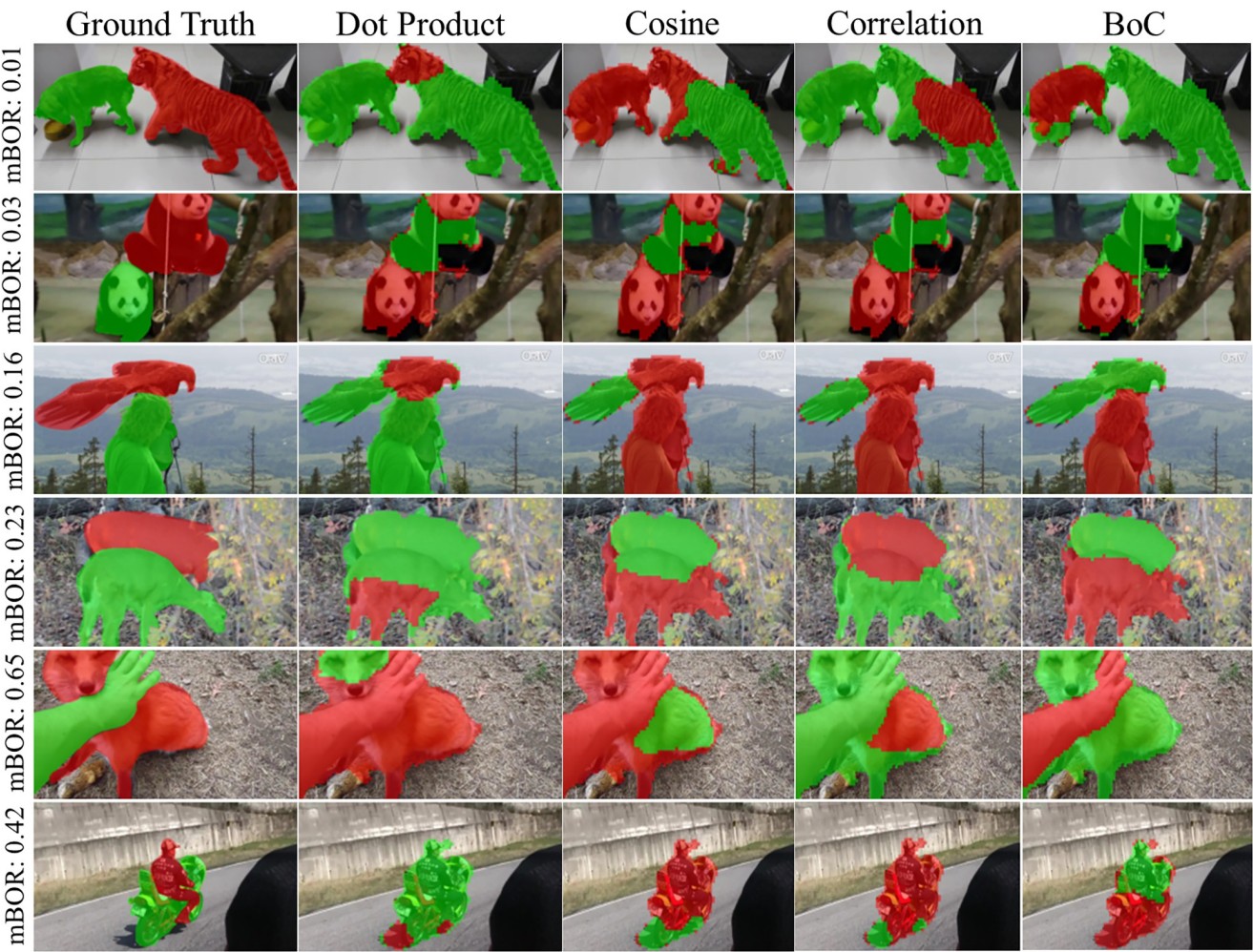

**Fig 11. Results of instance segmentation under various occlusion conditions.** Instance segmentation results under varying levels of occlusion, represented by the MBOR value while utilizing different metrics for creating the affinity matrix. The proposed metric, *BoC*, outperforms other metrics and produces more accurate masks, even in scenarios with heavy occlusions.

handles this situation more effectively. Table 4 and Fig 12 present the quantitative and qualitative results, where the ratio represents the sum of the ratios of each object's size to the size of the largest object.

Table 5 demonstrates the effect of the distance between instances on the segmentation process. When instances are closer together, the BoC metric performs better than other metrics. Another experiment, shown in Table 6, investigates the impact of the keypoint ratio between the Foreground and Background using the SIFT algorithm [55]. The smoothness value indicates the number of keypoints in the Foreground relative to the Background. A high smoothness value means the Foreground has more corner points than the Background, while a low value means the Background has more corner points. The BoC metric consistently outperforms other metrics by effectively accounting for the distribution of features, regardless of whether the Background or Foreground has more corner points. It's important to note that the number of images in all intervals in each of the Tables 3–6 is equal.

An experiment was conducted to validate the proposed BoC metric further. Firstly, 10 pixels were randomly sampled from each ground-truth mask. Subsequently, the intra-instance

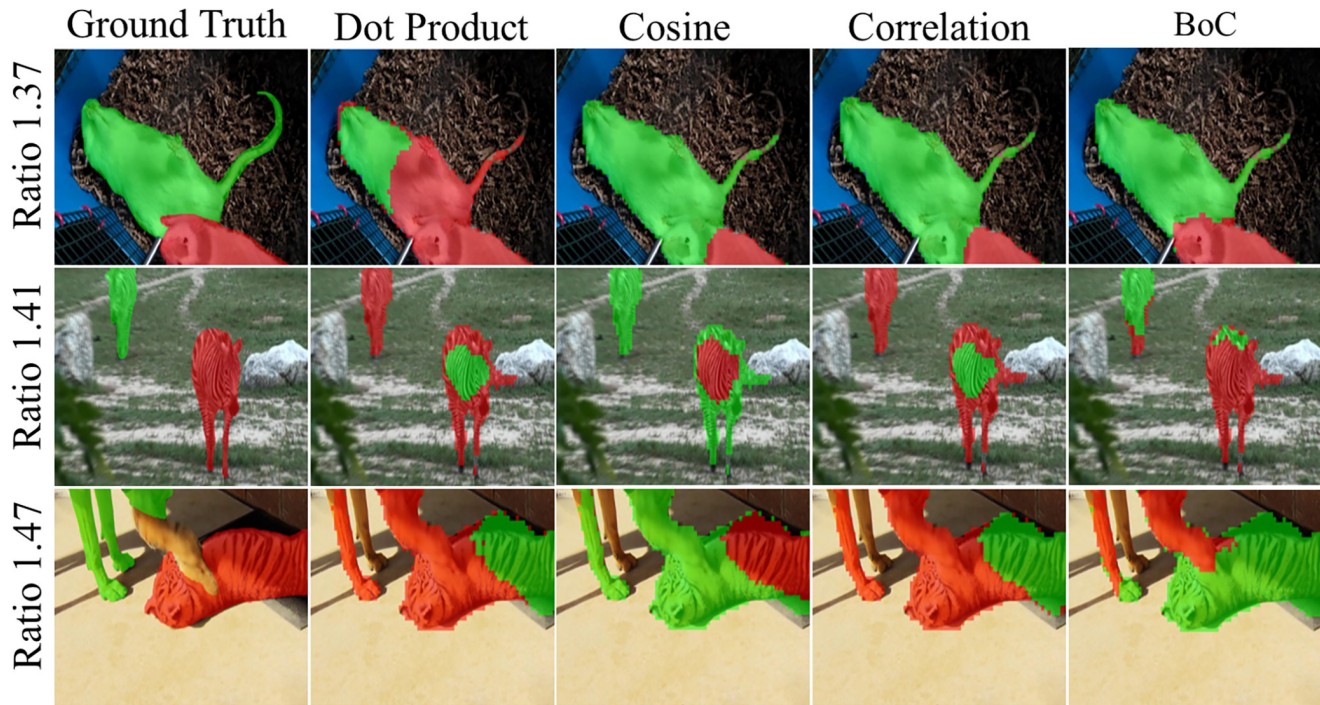

**Fig 12. Results of instance segmentation across different scale ratio levels.** Extracted masks for different metrics on the Youtube-VIS 2019 dataset. A lower value of ratio indicates greater variation in object sizes.

distances between these pixels were calculated using various similarity metrics. It was expected that these intra-instance distances from the same instance would exhibit low variance and similarity. The mean of intra-instance variance, $mVar_{intra}$, was computed. The mean of inter-instance distance variance, $mVar_{inter}$, was also calculated. The $mVar_{intra}$ to $mVar_{inter}$ ratio, denoted as $mR$, was compared for different similarity metrics. As shown in Fig 13, the proposed BoC metric demonstrated a lower mR value than other metrics. This result validates that the BoC metric better accounts for both intra-instance and inter-instance similarities compared to other metrics, resulting in improved instance segmentation.

Table 7 includes the ablation analysis for the proposed components. The results demonstrate that all components significantly contribute to improving the performance of instance

**Table 5. Quantitative results for different metrics under varying levels of distance between instances are presented in terms of mIoU on the Youtube-VIS 2019 dataset.**

| Metric | Distance <0.29 | Distance 0.29–0.41 | Distance ≥0.41 |
|---|---|---|---|
| Mahalanobis | 26.29 | 25.86 | 23.67 |
| L1 | 31.63 | 32.59 | 30.38 |
| Dot product | 33.12 | 33.51 | 31.50 |
| L2 | 32.68 | 33.63 | 31.99 |
| Chebyshev | 33.34 | 34.04 | 31.87 |
| Cosine | 33.27 | 34.66 | 32.76 |
| Correlation | 33.82 | 35.09 | 33.33 |
| Braycurtis | 33.85 | 35.27 | 33.31 |
| **BoC** | **34.25** | **35.58** | **33.39** |

**Table 6. Quantitative results for different metrics under varying levels of FG/BG smoothness in terms of mIoU on the Youtube-VIS 2019 dataset.**

| Metric | smoothness < 0.13 | smoothness 0.13–0.68 | smoothness ≥ 0.68 |
|---|---|---|---|
| Mahalanobis | 23.50 | 24.50 | 27.82 |
| L1 | 30.83 | 31.54 | 32.23 |
| Dot product | 32.46 | 32.35 | 33.32 |
| L2 | 32.32 | 32.76 | 33.23 |
| Chebyshev | 32.85 | 32.99 | 33.42 |
| Cosine | 32.95 | 33.95 | 33.79 |
| Correlation | 33.74 | 34.29 | 34.20 |
| Braycurtis | 33.52 | 34.31 | 34.59 |
| **BoC** | **33.99** | **34.47** | **34.76** |

segmentation. Starting from a baseline mIoU of 31.75%, the addition of NCR leads to a significant improvement to 32.92%, showcasing its individual effectiveness. Interestingly, introducing DCR alongside NCR results in a marginal decrease to 32.70%, suggesting a nuanced interaction between the two components that warrants further exploration.

Isolating BoC alone yields a lower mIoU of 30.50%, indicating that BoC's standalone contribution may not be as impactful in enhancing instance segmentation. However, combining NCR with BoC produces a synergistic effect, boosting mIoU to 33.62%. The most compelling outcome arises when all three components—NCR, DCR, and BoC—are integrated, achieving the highest mIoU of 34.41%. This holistic approach underscores the complementary nature of

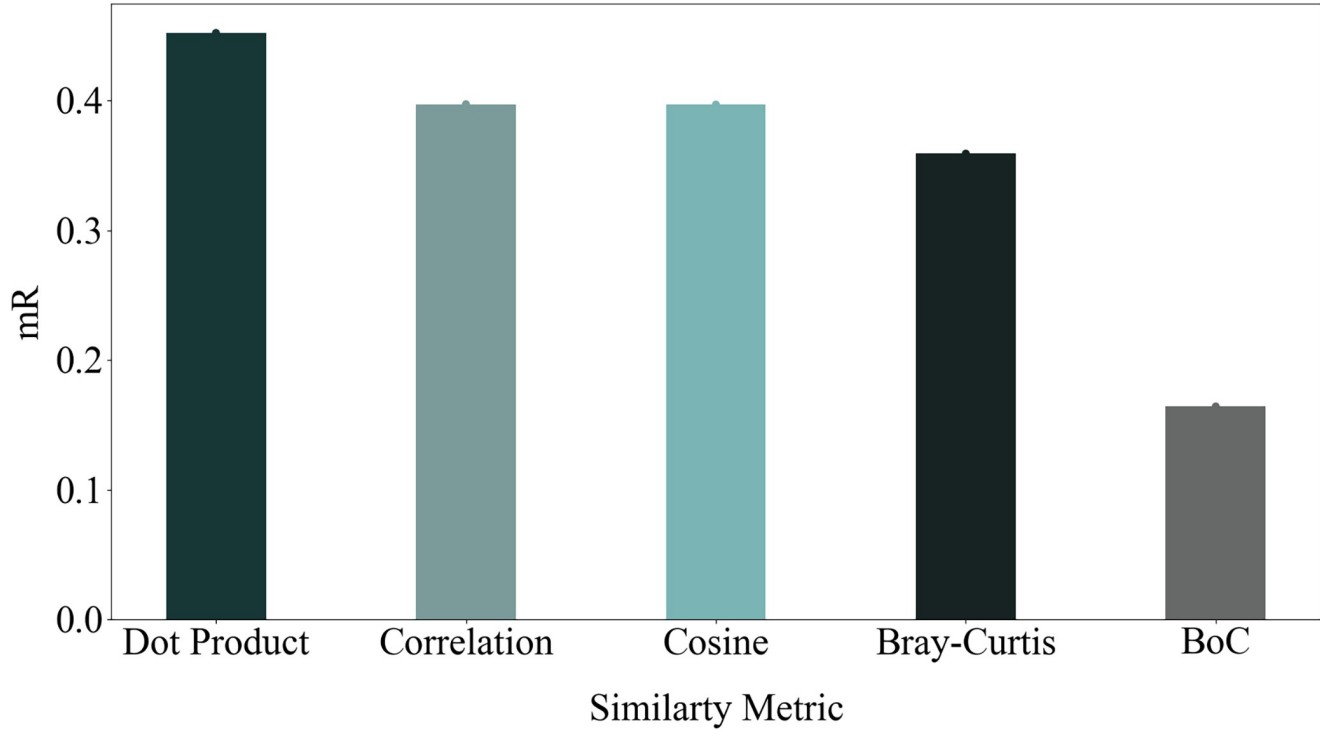

**Fig 13. Ratio of variance of intra-instance similarity to inter-instance similarity for different similarity metrics.** As depicted in this figure, the BoC metric showed a reduced mR value compared to alternative metrics. This finding confirms that the BoC metric captures intra-instance and inter-instance similarities more effectively than other metrics, thereby enhancing instance segmentation.

**Table 7. An ablation study to analyze the impact of proposed components on mIoU.**

| NCR | DCR | BoC | mIoU (%) |
|:---:|:---:|:---:|:---:|
| | | | 31.75 |
| ✓ | | | 32.92 |
| ✓ | ✓ | | 32.70 |
| | | ✓ | 30.50 |
| ✓ | | ✓ | 33.62 |
| ✓ | ✓ | ✓ | **34.41** |

the components, overcoming individual limitations and emphasizing their collective significance in advancing instance segmentation accuracy.

Fig 14 presents a visual analysis of the proposed BoC metric. As depicted, both the Chebyshev and Bray Curtis metrics exhibit poor performance in generating accurate instance masks. However, the BoC metric, which combines both the values and distribution of the features, produces superior instance masks compared to the conventional dot product.

## Conclusion

This paper focused on enhancing the performance of deep spectral methods specifically for instance segmentation purposes. To achieve this, two modules were proposed to retain channels with the most informative content from the feature maps obtained from a self-supervised backbone. Moreover, it was shown that the conventional use of dot product for creating the affinity matrix has limitations regarding instance segmentation. To address this issue, a novel

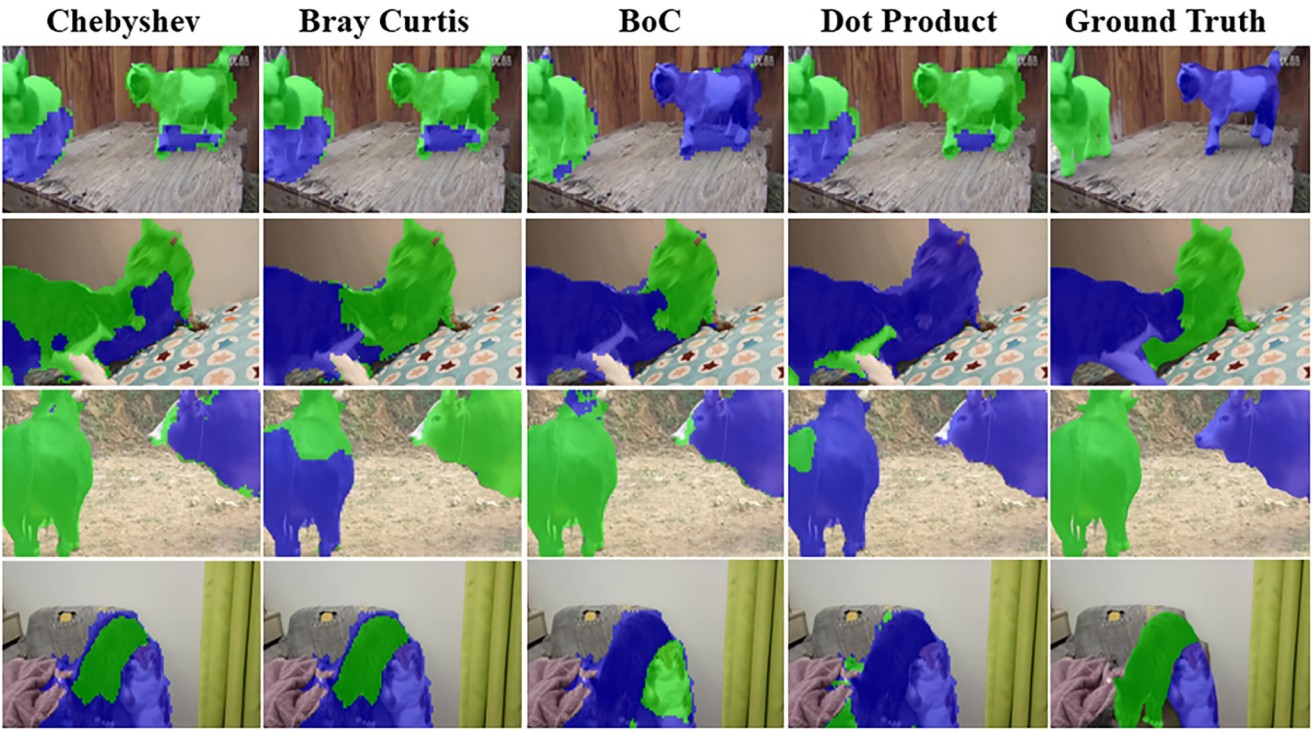

**Fig 14. An visual ablation study to analyze the impact of proposed BoC metric.**

similarity metric was introduced to improve the affinity matrix for instance segmentation tasks. The proposed components were designed to enhance the quality of eigensegments extracted through deep spectral methods for instance segmentation. These components can easily be integrated with any deep spectral method that aims to solve instance segmentation problems. It should be noted that in supervised learning, it would also be possible to train networks, specifically for both channel reduction modules, to retain the most informative channels effectively.

## Supporting information

**S1 Table.**
(PDF)

**S2 Table.**
(PDF)

**S3 Table.**
(PDF)

**S4 Table.**
(PDF)

**S5 Table.**
(PDF)

**S6 Table.**
(PDF)

**S7 Table.**
(PDF)

## Author Contributions

**Conceptualization:** Farnoosh Arefi, Shohreh Kasaei.

**Data curation:** Farnoosh Arefi.

**Formal analysis:** Farnoosh Arefi, Amir M. Mansourian, Shohreh Kasaei.

**Investigation:** Farnoosh Arefi, Shohreh Kasaei.

**Methodology:** Farnoosh Arefi, Amir M. Mansourian, Shohreh Kasaei.

**Project administration:** Shohreh Kasaei.

**Resources:** Farnoosh Arefi, Shohreh Kasaei.

**Software:** Farnoosh Arefi, Shohreh Kasaei.

**Supervision:** Shohreh Kasaei.

**Validation:** Farnoosh Arefi, Shohreh Kasaei.

**Visualization:** Farnoosh Arefi, Shohreh Kasaei.

**Writing – original draft:** Farnoosh Arefi, Amir M. Mansourian, Shohreh Kasaei.

**Writing – review & editing:** Farnoosh Arefi, Amir M. Mansourian, Shohreh Kasaei.

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
