## [Decision Letter · Decision Letter 0]

9 May 2024

PONE-D-24-05072Deep Spectral Improvement for Unsupervised Image Instance SegmentationPLOS ONE

Dear Dr. Kasaei,

Thank you for submitting your manuscript to PLOS ONE. After careful consideration, we feel that it has merit but does not fully meet PLOS ONE’s publication criteria as it currently stands. Therefore, we invite you to submit a revised version of the manuscript that addresses the points raised during the review process.

We look forward to receiving your revised manuscript.

Kind regards,

Yawen Lu,

Academic Editor

PLOS ONE

Journal Requirements:

4. Please ensure that you refer to Figure 11 and 12 in your text as, if accepted, production will need this reference to link the reader to the figure.

5. Please upload a copy of Supporting Information Figure/Table/etc. Supporting Information which you refer to in your text on page 15.

**Additional Editor Comments:**

PONE-D-24-05072

Dear Dr. Shohreh Kasaei:

I am writing to you regarding the above referenced manuscript that you submitted to Plos One.

Based on the enclosed two reviews, I am pleased to inform you that this manuscript is recommended for Major Revision in the journal.

Please carefully address the reviewers' comments and suggestions regarding figure captions, more qualitative results, literature reviews, etc., to improve the quality of the manuscript in the revised submission.

Reviewers' comments:

Reviewer's Responses to Questions

**Comments to the Author**

1. Is the manuscript technically sound, and do the data support the conclusions?

Reviewer #1: Yes

Reviewer #2: Yes

2. Has the statistical analysis been performed appropriately and rigorously? 

Reviewer #1: Yes

Reviewer #2: Yes

3. Have the authors made all data underlying the findings in their manuscript fully available?

Reviewer #1: Yes

Reviewer #2: Yes

4. Is the manuscript presented in an intelligible fashion and written in standard English?

Reviewer #1: Yes

Reviewer #2: Yes

5. Review Comments to the Author

Reviewer #1: The paper introduces a new method to improvise the performance of deep spectral methods specifically for instance segmentation. The authors incorporate two channel reduction modules to compare its effectiveness on the public datasets and validate the improvements in terms of mean IoU metric. The manuscript is complete and easily understandable, well experimented and contributes to new technical knowledge. I specifically liked the use of Bray-Curtis over Chebyshev instead of the conventional dot product for creating the affinity matrix. I suggest the manuscript to be accepted after a quick check for typographical errors and minor corrections stated below.

The following are minor corrections that can be incorporated to improve quality:

1) In the captions of figures 8 and 9, the ”M” has both closing quotes and at many other places, this is a common mistake while using LaTeX and can be resolved with using (`) character. Please review similar typo throughout the manuscript.

2) I urge authors if they could extend and incorporate the experiments to few other image segmentation datasets so it is accepted by wider audience. This further validates the findings and enhances the quality of the paper.

Reviewer #2: The article presents an interesting approach to improving deep spectral methods for the unsupervised instance segmentation task. The authors identify limitations in existing methods and propose novel techniques to address them. However, there are some aspects that should be improved.

1. The authors should strive for a more structured and logical flow in the proposed method section, ensuring the rationale and contributions are clear.

2. While the article includes some qualitative results, a more comprehensive visual analysis would greatly enhance the reader's understanding of the proposed method.

3. Although the authors have made efforts to include relevant literature in video and image segmentation, some are still missing in the related works section, such as [Coarse-to-fine video instance segmentation with factorized conditional appearance flows][Label-efficient video object segmentation with motion clues][Tube-Link: A Flexible Cross Tube Framework for Universal Video Segmentation][Tripartite feature enhanced pyramid network for dense prediction].

4. The writing quality and presentation of the article require improvement.

6. PLOS authors have the option to publish the peer review history of their article (what does this mean?). If published, this will include your full peer review and any attached files.

Reviewer #1: No

Reviewer #2: No

---

## [Author Response · Author response to Decision Letter 0]

27 May 2024

In relation to the review of the above mentioned manuscript, we would like to thank the respected anonymous reviewers for their valuable and constructive comments that led to improve the quality of the revised manuscript.

We are pleased to see that the impact of the research in the field of deep spectral methods and instance segmentation has been acknowledged by the reviewers (R1, R2). We appreciate that the manuscript has been recognized as well-written and well-experimented (R1), and we are glad that it is considered novel in addressing the limitations of existing methods (R2). Furthermore, we are grateful that the importance of the proposed Bray-Curtis over Chebyshev (BoC) metric has been acknowledged (R1). We are also delighted to hear that R1 has found the manuscript as ready for acceptance after addressing the minor reviews.

In the following, we have listed the issues raised by the respected reviewers and have addressed them as best as we could, and where appropriate we have revised the manuscript accordingly.

Reviewer 1

The paper introduces a new method to improvise the performance of deep spectral methods specifically for instance segmentation. The authors incorporate two channel reduction modules to compare its effectiveness on the public datasets and validate the improvements in terms of mean IoU metric. The manuscript is complete and easily understandable, well experimented and contributes to new technical knowledge. I specifically liked the use of Bray-Curtis over Chebyshev instead of the conventional dot product for creating the affinity matrix. I suggest the manuscript to be accepted after a quick check for typographical errors and minor corrections stated below.

1) In the captions of figures 8 and 9, the ”M” has both closing quotes and at many other places, this is a common mistake while using LaTeX and can be resolved with using (`) character. Please review similar typo throughout the manuscript.

We appreciate R1’s careful attention. It is edited in the revised version of the manuscript.

2) I urge authors if they could extend and incorporate the experiments to few other image segmentation datasets so it is accepted by wider audience. This further validates the findings and enhances the quality of the paper.

We highly value the suggestion provided by R1 and have taken it into consideration by incorporating two additional datasets into our evaluation. For the Foreground-Background (Fg-Bg) segmentation task, we have extended the testing to include the Densely Annotated VIdeo Segmentation (DAVIS 2016) dataset, which is widely recognized as a challenging benchmark in the field. Moreover, for the instance segmentation task, we have reported our results on the Occluded Video Instance Segmentation (OVIS 2022) dataset. This newly proposed benchmark is of great importance due to its occluded nature, which introduces additional challenges for accurate instance segmentation. Thanks to R1, we believe that reporting the results on these two challenging datasets has led to better show the effectiveness of the proposed method.

Reviewer 2

The article presents an interesting approach to improving deep spectral methods for the unsupervised instance segmentation task. The authors identify limitations in existing methods and propose novel techniques to address them. However, there are some aspects that should be improved.

1) The authors should strive for a more structured and logical flow in the proposed method section, ensuring the rationale and contributions are clear.

We apologize if the previous text was unclear. We have made changes in the revised version to address this issue by providing additional explanations where necessary and removing the redundant parts. Furthermore, we have incorporated a new paragraph at the beginning of the proposed method section, which outlines the logical flow of our approach. This addition aims to assist readers in understanding the proper sequence of the proposed method. We are grateful to R2 for providing this feedback that let to improving the clarity of the manuscript.

2) While the article includes some qualitative results, a more comprehensive visual analysis would greatly enhance the reader's understanding of the proposed method.

We completely understand R2's concern. In response, we have included a new figure (Figure 14), that showcases the outputs of the proposed method on the images from the newly added OVIS dataset. This contains an ablation study of the proposed BoC metric. It compares the outputs of the Bray Curtis, Chebyshev, and BoC metrics with that of the dot product and ground truth. The purpose of this visual comparison is to demonstrate the significance and necessity of the proposed BoC metric. Additionally, we have provided corresponding explanations for this figure in the Experimental Results section. We firmly believe that this additional visual analysis significantly enhances the reader's understanding of the underlying concepts and ideas behind the proposed method. 

3) Although the authors have made efforts to include relevant literature in video and image segmentation, some are still missing in the related works section, such as [Coarse-to-fine video instance segmentation with factorized conditional appearance flows][Label-efficient video object segmentation with motion clues][Tube-Link: A Flexible Cross Tube Framework for Universal Video Segmentation][Tripartite feature enhanced pyramid network for dense prediction].

We express our gratitude to R2 for providing this detailed feedback. We have thoroughly explored all the mentioned papers, as they are influential works recently published in top venues. Following careful examination, we have identified the most relevant papers that align with this research and have appropriately cited them in the relevant sections of the revised version. 

4) The writing quality and presentation of the article require improvement.

Thanks to R2, we have dedicated significant effort to revising the paper for multiple times with the aim of enhancing its readability. Additionally, we sought input from colleagues, who reviewed the paper and provided feedback on any ambiguous sections. As a result of these revisions and consultations, we believe that the revised version has significantly improved in terms of writing quality and presentation. We hope that the concerns raised by R2 have been effectively addressed.

---

## [Decision Letter · Decision Letter 1]

17 Jun 2024

Deep Spectral Improvement for Unsupervised Image Instance Segmentation

PONE-D-24-05072R1

Dear Dr. Shohreh Kasaei,

We’re pleased to inform you that your manuscript has been judged scientifically suitable for publication and will be formally accepted for publication once it meets all outstanding technical requirements.

Kind regards,

Yawen Lu, Ph.D

Academic Editor

PLOS ONE

Additional Editor Comments (optional):

Dear authors:

Regarding your submission:

PONE-D-24-05072R1

Deep Spectral Improvement for Unsupervised Image Instance Segmentation

We have received feedbacks from the previous reviewers and are announcing that your work has been Accepted for publication in PLOS ONE.

Please follow the following steps and provide a camera-ready version of your manuscript. Congratulation!

Reviewers' comments:

Reviewer's Responses to Questions

**Comments to the Author**

1. If the authors have adequately addressed your comments raised in a previous round of review and you feel that this manuscript is now acceptable for publication, you may indicate that here to bypass the “Comments to the Author” section, enter your conflict of interest statement in the “Confidential to Editor” section, and submit your "Accept" recommendation.

Reviewer #1: All comments have been addressed

2. Is the manuscript technically sound, and do the data support the conclusions?

Reviewer #1: Yes

3. Has the statistical analysis been performed appropriately and rigorously? 

Reviewer #1: Yes

4. Have the authors made all data underlying the findings in their manuscript fully available?

Reviewer #1: Yes

5. Is the manuscript presented in an intelligible fashion and written in standard English?

Reviewer #1: Yes

6. Review Comments to the Author

Reviewer #1: The Authors have addressed all the comments including experiments with additional datasets and the manuscript can be accepted in present form.

7. PLOS authors have the option to publish the peer review history of their article (what does this mean?). If published, this will include your full peer review and any attached files.

Reviewer #1: No

---

## [Editor Report · Acceptance letter]

10 Jul 2024

PONE-D-24-05072R1 

PLOS ONE

Dear Dr. Kasaei, 

I'm pleased to inform you that your manuscript has been deemed suitable for publication in PLOS ONE. Congratulations! Your manuscript is now being handed over to our production team.

Kind regards, 

on behalf of

Dr. Yawen Lu 

Academic Editor

PLOS ONE